# Hybrid molecular graphene transistor as an operando and optoelectronic platform

Jorge Trasobares [1,2] ✉, Juan Carlos Martín-Romano [1], Muhammad Waqas Khaliq[3,4], Sandra Ruiz-Gómez [3], Michael Foerster[3], Miguel Ángel Niño[3], Patricia Pedraz [1], Yannick. J. Dappe[5], Marina Calero de Ory [6], Julia García-Pérez[1], María Acebrón[1], Manuel Rodríguez Osorio[1], María Teresa Magaz[5], Alicia Gomez [5], Rodolfo Miranda [6,7] & Daniel Granados [1] ✉

Lack of reproducibility hampers molecular devices integration into large-scale circuits. Thus, incorporating operando characterization can facilitate the understanding of multiple features producing disparities in different devices. In this work, we report the realization of hybrid molecular graphene field effect transistors (m-GFETs) based on 11-(Ferrocenyl)undecanethiol (FcC$_{11}$SH) micro self-assembled monolayers (μSAMs) and high-quality graphene (Gr) in a back-gated configuration. On the one hand, Gr enables redox electron transfer, avoids molecular degradation and permits operando spectroscopy. On the other hand, molecular electrode decoration shifts the Gr Dirac point (V$_{DP}$) to neutrality and generates a photocurrent in the Gr electron conduction regime. Benefitting from this heterogeneous response, the m-GFETs can implement optoelectronic AND/OR logic functions. Our approach represents a step forward in the field of molecular scale electronics with implications in sensing and computing based on sustainable chemicals.

The field of molecular electronics[1] opens the door to exploring quantum phenomena for computing[2,3]. Indeed, properly designed molecules offer complimentary design to inorganic materials. As a result, the field has been intensely reviewed[4,5]. Unfortunately, there is lack of reproducibility when measuring tunneling rates across the same type of molecules. Differences in the resistance per molecule measured through molecular junctions can be up to 8 orders of magnitude. This feature is often attributed to variances in contacts, surface roughness, presence of defects and how all these factors scale with the junction size[6]. Then, establishing a robust[7] and industrially scalable platform is highly desirable. Three fundamental aspects must be observed: the molecular bridge, the interface and the electrode material[8]. The molecular bridge energy level may be tuned with redox molecules[9].

Its fundamental mechanism is founded in the ability to switch between molecular energy levels or oxidation states. Electrons can be localized in the redox state during the transmission process depending on the relative alignment of molecular energy and the electrode Fermi level. A well-studied redox group is ferrocene (Fc), in particular, the commercially available FcC$_{11}$SH which is an archetype molecule broadly explored in electrochemistry[10,11] and molecular electronics[12,13]. Researchers have investigated the impact of molecular length[14], its performance at high frequency[15] and operation in flexible devices[13]. Secondly, continuous and pristine interface produced by the molecular decoration of the electrodes with self-assembled monolayers (SAMs) can tune the electrode work function with high impact on the performance of organic optoelectronic devices[16]. Thirdly, Gr is an

[1]IMDEA-Nanociencia, Cantoblanco, Madrid 28049, Spain. [2]Department of Biodiversity, Ecology and Evolution (Biomathematics), Universidad Complutense de Madrid, Madrid 28040, Spain. [3]ALBA Synchrotron, Carrer de la llum 2-26, Cerdanyola del Valles 08290, Spain. [4]Department of Condensed Matter Physics, University of Barcelona, Barcelona, Spain. [5]Centro de Astrobiología (CSIC-INTA), Torrejón de Ardoz 28850, Spain. [6]SPEC, CEA, CNRS Université Paris-Saclay, Gif-sur-Yvette 91191, France. [7]Dpto. de Física de la Materia Condensada, Universidad Autónoma de Madrid, Cantoblanco, Spain. ✉e-mail: jtrasoba@ucm.es; Daniel.granados@imdea.org

excellent 2D material for the construction of transistors[17] and a perfect electrode candidate to link the macro (electrode) and the nano (molecule) domains. In particular, π-π interactions between Fc and Gr are ideal for electrical connections and a transparent electrode gives access to interact with light. On the one hand, optical spectroscopies are convenient to analyze the molecular bridge between leads[18]. On the other hand, light is a control knob to tune the conduction properties by inducing photophysical or photochemical changes in the structure[19,20]. In this work, we first investigate the properties of a single SAM//Gr hybrid junction and then integrate the same material in an optoelectronic platform (m-GFET) based on Gr and micro self-assembled monolayers (μSAMs) made with FcC$_{11}$SH. Our design, free of ionic liquid[21], uses Gr as a soft, high conductive and transparent top contact that encapsulates the molecular entity of interest, allowing both operando characterization and optical gating.

## Results and discussion
### Graphene top electrode at the single junction level

We first describe the large area ferrocenyl junctions both uncovered (Au/SC$_{11}$Fc) and covered (Au/SC$_{11}$Fc//Gr) by a Gr layer, and then use this material to construct the m-GFET device. Note that "/" and "//" refer to chemical and non-chemical bonds respectively. A schematic representation of the uncovered junction including main parameters is displayed in Fig. 1a, while Fig. 1b represents the hybrid Au/SC$_{11}$Fc//Gr junction where Gr offers a transparent and conductive barrier between the SAM and the environment. Electron transfer is explored through Cyclic Voltammetry (CV). In Fig. 1c, the electrochemical response of Au/SC$_{11}$Fc SAMs (blue line) shows a conventional redox couple. Our experimental data can be fitted using the extended Laviron model[22] and a background (step model) related to the admittance of the double-

layer capacitor generated during the process, recently exploited to reach attofarad sensitivity[23] (Eq. 1 in Theoretical Methods and Supplementary Note 1). The oxidation potential appears at 280 mV with respect to the silver chloride reference electrode (Ag/AgCl). We extract a molecular energy level $E_H \approx -5$ eV (Eq. 2 in Theoretical Methods). The integration of the oxidation (or reduction) wave results in a charge amount that relates to the surface coverage, $\Gamma = 4.1 \pm 0.5 \times 10^{-10}$ mol/cm² (Eq. 3 in Theoretical Methods and Supplementary Table 1). We find a $FWHM = 94$ mV resulting in slightly repulsive interactions (Eq. 4 in Theoretical Methods). Remarkably, Au/SC$_{11}$Fc//Gr SAMs continue to exhibit the oxidation-reduction process. Electron transfer is permitted under the Gr layer but there exists a drop in the current (orange line). This drop can be attributed to the presence of multilayer domains in the Gr[21] (Supplementary Note 2 for optical and Raman characterization of CVD Gr layers) or a slightly deficient contact in some regions of the interface between the Fc group and the Gr top electrode[24]. Another significant approach is obtaining XPS spectra of a SAM under the graphene layer. Oxidation and ageing effects of the uncovered and covered structures are analyzed by X-ray photoelectron spectroscopy (XPS). Figure 1d presents the XPS spectra of the Fe 2p core level energies. In the fresh Au/SC$_{11}$Fc and Au/SC$_{11}$Fc//Gr SAMs Fe doublets located at 707.6 eV and 720.4 eV are clearly observed. This doublet corresponds with the Fe 2p$_{3/2}$ and Fe 2p$_{1/2}$ respectively. We appreciate a slight decrease in the height of the 707.06 eV peak on the Gr covered samples but not a binding energy shift, invalidating charge transfer between Fc and Gr. After sixty days of ageing under atmospheric conditions, the aged Au/SC$_{11}$Fc//Gr SAM does not show a substantial change in the spectrum, while the uncovered aged sample presents clear signatures of oxidized species. Ferrocenium peaks at 710.2 eV and 723.5 eV appear in this situation. More details are given in Supplementary Note 3.

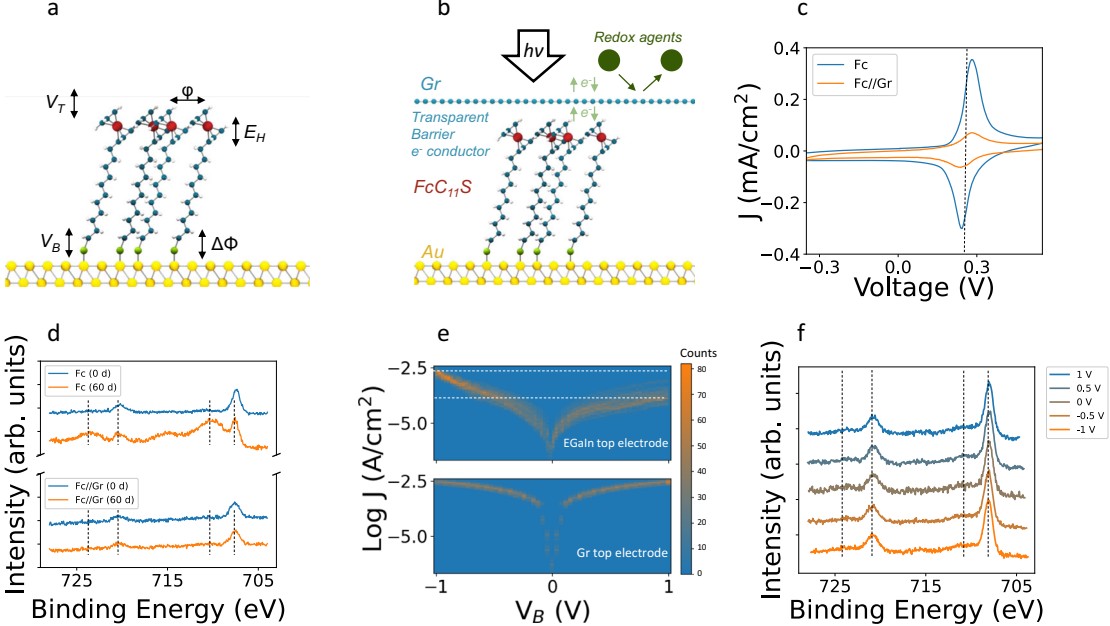

**Fig. 1 | Electron transfer and transport at the junction. a** Schematic representation of an uncovered junction Au/SC$_{11}$Fc SAM including main parameters: molecular energy ($E_H$), electronic coupling with the bottom electrode and a possible top one ($V_B$ and $V_T$), intermolecular coulombic interactions ($\varphi$) and work function change ($\Delta\Phi$). **b** Schematic representation of the hybrid SAMs (Au/SC$_{11}$Fc//Gr) including the additional characteristics of a transparent conductive barrier. Atomic details of SAMs are symbolized by spheres of different colors and sizes: Fe (red), Au (yellow), S (green), C (blue) and H (white). hv represents the incident light or beam for optical gating and/or spectroscopic characterization. e- indicates electron transfer during redox process. **c** Electrochemical characterization of Au/SC$_{11}$Fc (blue) and Au/SC$_{11}$Fc//Gr (orange) SAMs. CV is recorded at a

6 mm in diameter gold working electrodes at 1 V/s versus Ag/AgCl reference electrode, 1 M NaClO$_4$ electrolyte and a silver bar counter electrode. $J$ represents current density measured in mAcm⁻². Dotted line indicates the formal potential of the Fc group[30]. **d** Fe 2p XPS of the fresh (blue) and aged (orange) Au/SC$_{11}$Fc and Au/SC$_{11}$Fc//Gr samples (intensity in arbitrary units is displaced on "y" scale for clarity). **e** 2D histogram of the Log $J(V)$ taken from 120 current versus voltage scans for Au/SC$_{11}$Fc//EGaIn and Au/SC$_{11}$Fc//Gr junctions. **f** Operando Fe 2p XPS of Au/SC$_{11}$Fc//Gr at 1, 0.5, 0, −0.5 and −1 V. In **d** and **f** Fe doublets located at 707.6 eV and 720.4 eV and Ferrocenium peaks at 710.2 eV and 723.5 eV are highlighted by vertical dashed lines for guide the eye.

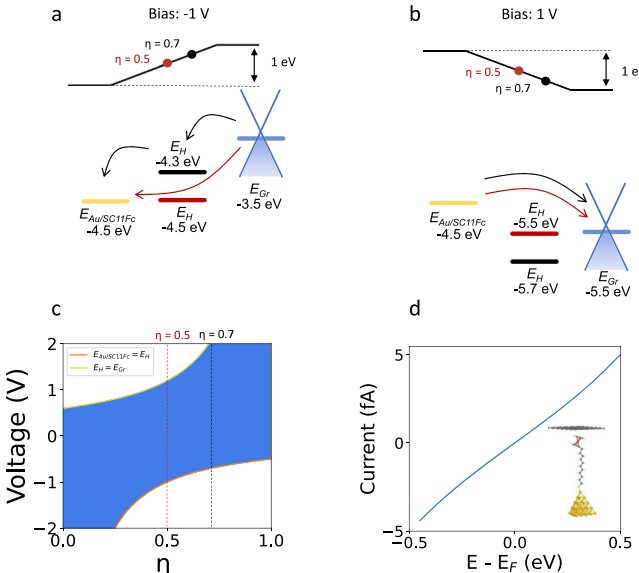

**Fig. 2 | Electron transport modeling at the junction.** Band diagram illustrations at (**a**) −1 V and (**b**) +1 V bias. Fermi levels of electrodes, $E_{Au/SC11Fc}$ and $E_{Gr}$, are calculated from the work functions obtained by UPS experiments in Supplementary Fig. 8 ($\Phi_{Gr}$ = 4.5 eV, $\Phi_{Au/SC11Fc}$ = 4.5 eV). Molecular energy level, $E_H$ = −5 eV, is found from cyclic voltammetry experiments and Eq. 2. Voltage drop within the junction, is represented by $\eta$. It is both set to $\eta$ = 0.7 as standard value for EGaIn top electrode[6] and $\eta$ = 0.5 as a value explaining the loss of rectification with graphene top electrode. **c** Operation regions for molecular diodes connected with Au and Gr electrodes as a function of the voltage drop within the junction. White and blue backgrounds represent forward and reverse regimes respectively. **d** DFT calculated I (V) curve for a single Au/SC₁₁Fc//Gr molecular junction. Inset: Schematic representation of the calculated single molecular junction.

## Theoretical results at the single molecule level

We have theoretically modeled our hybrid junction to understand the experimental results shown in the preceding section. From the energy

Afterwards, electron transport is first characterized with the well-known EGaIn technique as a starting point benchmark with previous work[14,25]. Figure 1e displays the log current density versus applied voltage between the cone-shaped EGaIn tip and the uncovered SAM on top of the Au electrode. A rectification ratio slightly higher than one order of magnitude for the recommended contact area of ~500 μm² is obtained[6]. Current densities span from Log J (Acm⁻²) = −3.7 at +1 V to −2.5 at −1 V (reverse and forward bias respectively). As first proposed by C. A. Nijhuis et al. the rectification in these 11-(ferrocenyl)−1-undecanethiol junctions[25–27] emerges from two different transport mechanisms in reverse (bond tunneling) vs forward (sequential tunneling) polarization[13,15,26]. Usually, temperature-dependent measurements are applied to distinguish those regimes[27]. Our graphene covering 11-(ferrocenyl)−1-undecanethiol hybrid SAMs can overpass this tedious temperature-dependent characterization or limitations by enabling operando spectroscopy. For example, our approach may present advantages in junctions pushed to work in the inverted Marcus Regime where the transport mechanism becomes activationless[28]. Hence, we replace the top opaque EGaIn to a transparent Gr. In these hybrid junctions, we observe a suppression of the electrical rectification and a reduction in the current dispersion (bottom part of Fig. 1e). Both aspects might be the effects of the top electrode or the large area (mm²) required to accomplish XPS at our in-home facilities. Figure 1f shows the operando XPS biasing the Gr at +1, 0.5, 0, −0.5 and −1 V. In all cases we observe the spectrum finger print of the Fc group during the electrical operation[29] and a rather constant signature of the oxide components.

level diagram perspective, both Gr and EGaIn top electrodes should give similar rectification ratios[25,26] due to the close proximity of the Fermi level energies (4.5 eV)[31]. It is essential to note that the performance in the forward and reverse regimes depends on the working function of the electrodes and the voltage drop. If the molecular energy level falls between the Fermi level of the electrodes, the diode will operate in forward, otherwise in reverse. Based on the Ultraviolet photoelectron spectroscopy (UPS) (Supplementary Fig. 8) we obtain a working function of the decorated electrode of $\Phi_{Au/SC11Fc}$ = 4.5 eV, then we consider that the only free parameter to explain the lack of rectification with the Gr top electrode is the voltage drop ($\eta$)[32]. Figure 2a presents two different electron transport mechanisms for $\eta$ equal to 0.7 and 0.5. For a $\eta$ = 0.7, two black arrows represent the sequential tunneling in the forward regime. It could explain the electrical response with the EGaIn electrode. However, a $\eta$ = 0.5 moves the molecular energy level away from the Fermi level window of the electrodes, providing a direct tunneling mechanism within the junction. That explains the electrical characteristics of the Gr top electrode. Furthermore, our operando spectroscopy does not detect the operation in the forward regime where the contribution of the oxidized species should appear. Figure 2b presents in both cases ($\eta$ = 0.7 and $\eta$ = 0.5) a direct tunneling mechanism. Figure 2c, with the orange line matching the first part of the inequality and the yellow line, the second. The white and blue background shading represents forward and reverse regimes. As shown in Fig. 2c, voltage drop within the molecular junction[33] is a critical parameter to pay special attention. Particularly, for large voltage windows[34] where, without a careful consideration, it may appear a forward regime producing the breakdown before reaching 2 V. Furthermore, we model a single molecular junction, following a well-established methodology[35]. Within a Density Functional Theory (DFT) formalism, we determine the electronic structure and in particular the projected Density of States (PDOS) of the molecule between the gold and Gr electrodes. The corresponding atomic configuration is represented in the inset of Fig. 2d. A quasi-symmetric distribution of the electronic levels near the Fermi level appears at around −0.5 and +0.35 eV (Supplementary Note 5). In Fig. 2d we integrate the electronic transmission for several biases taking properly into account the couplings between electrodes to determine a theoretical I(V) curve. As experimentally observed, the model provides a very low rectification characteristic, most probably originating from the symmetry breaking induced by the Gr electrode. This has indeed been observed in previous works[35,36], where it was shown that the introduction of a Gr electrode at one side of the junction breaks the symmetry of the system and leads to a reduced attenuation factor compared to the one obtained in the usual gold-gold symmetric junction.

## Construction of the m-GFETs platform

With all the previous information, we now implement this Fc//Gr hybrid material in a field effect transistor configuration. This section is devoted to the m-GFET platform (Fig. 3, Experimental Methods and Supplementary Note 6). The fabrication process begins with the definition of the two source-drain electrodes made of Cr/Au bi-layered (6/60 nm respectively) via conventional lithography and lift-off process. The electrodes are deposited on top of a 290 nm thick layer of silicon oxide (SiO₂) on a highly n++ doped silicon substrate. Electrodes are electrically isolated by a thin conformal coating of HfO₂ (~8 nm) grown by atomic layer deposition (ALD). At the end of each electrode, a circular area of 5 μm in diameter is etched by hydrofluoric acid (HF) to expose the Au surface. Next, conductive atomic force microscopy (CAFM) confirms an effective isolation of the HfO₂ layer and ohmic contact through the etched pore to the Au electrode. The Au exposed areas in the source and drain electrodes are then decorated with SC₁₁Fc μSAMs formed by conventional thiol adsorption, giving rise to two uncovered junctions on each electrode. CVD Gr is transferred on top of

the electrodes bridging the source and drain to form two covered junctions. A key aspect is the ability of Gr to adapt to the surface topography of the exposed micropore in the range of 2–10 nm[37,38] and

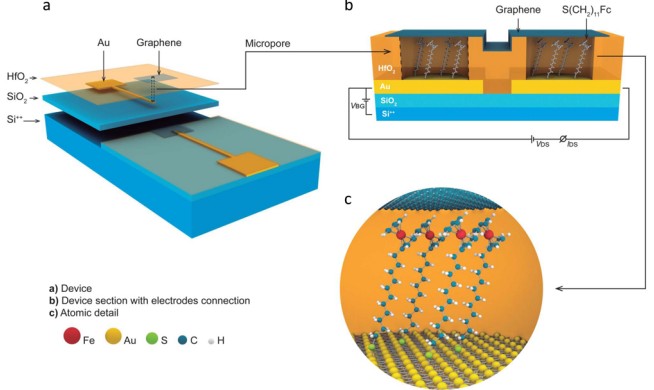

**Fig. 3 | Schematic illustration of the m-GFET. a** Design of the fabricated device where the different layers are artificially separated for clarity. Fabrication is supported on a Si$^{++}$/SiO$_2$ (290 nm) wafer. Starting with a photolithography step for the fabrication of the Cr/Au (6/60 nm) electrodes, followed by 8 nm ALD growth HfO$_2$ to insulate the electrodes and opening of micropores in the HfO$_2$ by dry etching (HF). The micropores areas allow controlling the surface area of the electrical contact. After thiol adsorption and Gr transfer, a third lithography step to pattern the Gr completes the fabrication process. **b** Cross-section of the device with electrical connections including: back gate voltage ($V_{BG}$), drain source voltage ($V_{DS}$) and drain source current ($I_{DS}$). **c** Molecular detail. Sulfur and gold are linked by a covalent bond, upwards 11 sp$_3$ carbon atoms and on top the Fc group connecting a single Gr layer by π-π interactions. Drawing is not to scale.

contact the μSAM by π–π interaction with the Fc group. Note that the homogeneous HfO$_2$ film is 8 nm thick and the μSAM is 2.4 nm. Since the rest of the Au is passivated with HfO$_2$, the Gr only bridges the two μSAMs. Finally, the top contact Gr is patterned by an extra lithographic step with no damage to the molecular properties due to the Gr barrier protection. Using this protocol, we have an operating yield of 65%.

## Operando charge transport in the ferrocenyl m-GFETs

The Dirac point is a singular characteristic of Gr-based transistors. The applied gate voltage ($V_{BG}$) at which the Dirac point is crossed ($V_{DP}$) varies with several factors, such as the work function of the electrodes or charges trapped at different interfaces. If we deposit our Gr on top of the electrodes in a field-effect transistor configuration (in the absence of the FcC$_{11}$SH, GFET), a significant charge transfer between Gr and the exposed electrode takes place, producing a voltage drop and subsequently a shift in the Gr Fermi level. We observe shifts of the Dirac point up to $V_{DP} = 150$ V (blue line in Fig. 4a and Supplementary Note 7). However, in the m-GFET, due to the molecular functionalization of the electrodes that tune their working function, the Dirac point remains close to 0 V (orange curve in Fig. 4a). The increment in energy is proportional to the change of the working function $\Delta E \approx \Delta \Phi$ and can be calculated by the shift on the neutrality point following Eq. 7 in Theoretical Methods[39–41]. The observed difference in the Dirac point of $\Delta V_{DP} = 150$ V relates to a variation in the working function of the electrodes of $\Delta \Phi \approx 0.5$ eV, which is in agreement with our UPS experiments (Supplementary Fig. 8) and earlier reported SAMs[16] but slightly higher than those considered for ferrocenyl SAMs[42]. Supra-molecular interactions, ion movement and charge trapping effects related to device geometry and fabrication conditions can alter the electronic response of the device. We observe well-known hysteresis loops[43] and

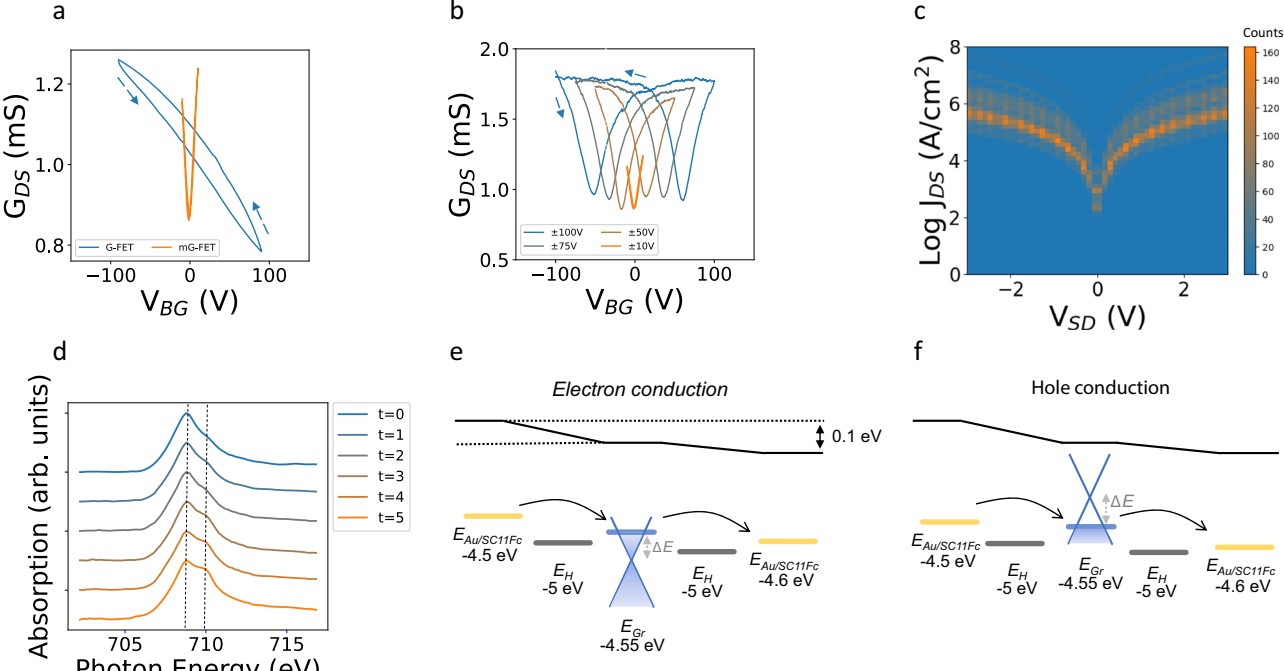

**Fig. 4 | Operando charge transport in the m-GFET configuration. a** Conductance versus gate voltage for both GFET and m-GFET devices at ambient conditions. **b** Conductance versus gate voltage for m-GFET devices at ambient conditions and different gate voltages windows. The arrow indicates the gate voltage sweeping direction. **c** Log $J_{DS}$ current at $V_{BG} = 0$, 2D map of a m-GFET over 120 sweeps (15 curves in 8 different devices). **d** Evolution of the X-ray absorption spectra from a 5 μm pore where the FcC$_{11}$SH is placed and connected with graphene on top at different source to drain voltages, measurements from t = 0 to t = 5 are performed every 20 minutes by changing the V$_{DS}$ from 0 to 1, 2, 0, −1 and −2 V. Vertical dashed lines are incorporated for guide the eye at 708.8 and 710 eV that corresponds to the presence of Fe$^{2+}$ and Fe$^{3+}$ respectively. **e, f** Display the schematic illustration of the transport model during electron and hole conduction. Considering a gold Fermi level of $E_{Au} = 5$ eV and a change of $\Delta \Phi = 0.5$ eV when decorating the electrodes the resulting Fermi level of the molecularly functionalized electrode is $E_{Au/SC11Fc} = -4.5$ eV. $V_{DS}$ is low enough to keep $E_H$ outside the $E_{Au/SC11Fc} - E_{Gr}$ window in both junctions of the m-GFET. $\Delta E$ is the energy difference between the Fermi level at a given $V_{BG}$ and the graphene Dirac point on the device.

alterations in the $V_{DP}$ as a function of the $V_{BG}$ window, scan rate and sweep direction. Effects are most probably produced by charges trapped in the device. On the m-GFETs, when the $V_{BG}$ is swept at a short range (±10 V) the hysteresis is diminished. However, when we increase the $V_{GB}$ the hysteresis spreads. It is an indication that charges are injected into the bulk SiO$_2$ or the interface[43,44]. The number of trapped charges (N) are calculated with Eq. 8 in Methods and displayed in Supplementary Table 2. When the $V_{BG}$ is swept ±100 V there is a Dirac point shift 55 V. This is equivalent to a trapped charge density[17] of $2 \times 10^{12}$ cm$^{-2}$. Although in many applications trapped charges may be, undesirables and attempted to minimize[45,46], this hysteretic mechanism may be rich for the purpose of nonvolatile memory devices and give extra function in chemical or biological sensing. The mobility measured with the transconductance values varies from 500 to 1000 cm$^2$V$^{-1}$s$^{-1}$. In Fig. 4c, we show the log $J$ vs. $V_{DS}$. In this case, the current needs to flow twice through the molecules, one from the thiol to the Fc and another time in the opposite direction. Consequently, despite the performance of the molecular diodes with Gr as a top electrode, there is no rectification in the current flowing from source to drain in our m-GFET. In this configuration we mainly measure the resistance of the reverse diode (Fig. 4c, Supplementary Note 8). We obtain a current density of $10^5$ A/cm$^2$ at 1 V in the range of direct evaporated or Gr devices[47]. In order to clearly discuss the electrochemical processes occurring inside the junction, a tremendous advantage of our design is to be able to perform Photoemission electron microscopy (PEEM) following the state of the SAM while the transistor is operating. Laterally resolved X-ray absorption spectroscopy (XAS) performed at the ALBA synchrotron radiation facility with a PEEM microscope shows in the electrode pore a clear Fe signal at the Fe L3 edge during all the experiment (Supplementary Note 9 and Fig. 4d). XAS spectra confirm the grafting of the ferrocenyl μSAMs only at the Au exposed pore under Gr. In contrast, to our results in Fig. 1c and other configurations where authors demonstrate chemical and electrochemical ferrocene oxidation and reduction by applying a voltage to an electrolyte on top of the graphene[21], the oxidation state of Fc is independent of the source-drain voltage remaining in the reduction state. Similar to other XAS synchrotron radiation experiments[48], there is photochemical switching due to the intense photon irradiation, common in other molecular systems. In our μSAMs composed of Au/SC$_{11}$Fc//Gr we observe an oxidation evolution from Fe$^{2+}$ (708.8 eV) to Fe$^{3+}$ (710 eV) in contrast to the reduction evolution reported for Au/SC$_{11}$Fc exposed to ultra-high vacuum[48]. Differences with respect to the work of Fan Zheng et al. are the micrometer size of the SAMs, presence of the Gr covering layer and operando conditions. In view of the above, the usual operation of these transistors is performed at a low voltage of 0.1 V. Where according to the band diagram for electron and hole-conduction regimes (found in Fig. 4e, f) show that the molecular energy level would not come into play. Then, electron transport would be tunneling through the entire molecule. Additionally, threshold PEEM measurements were performed to obtain a surface potential map of the m-GFETs. In Supplementary Fig. 26 can be seen how the spectra are shifted when applying different $V_{DS}$ (−1,0 and +1 V), this shift is related to the voltage drop.

## Photo-induced charge transfer

Finally, this section discusses results of photon absorption by the Fc group and subsequent electron transfer to the Gr channel. The Fc group is expected to absorb light in the UV-visible range, presenting two absorption bands at 322 nm and 442 nm[49]. In our m-GFET device, due to the long dynamics associated with charge-trapping under ambient experiments previously discussed, a shift in the Dirac point between dark and under illumination operation is challenging to appreciate[50]. However, pulsed light strongly influences the source to drain current only in the "electron conductivity" regime of the top layer Gr (Supplementary Note 10 of supplementary information). In Fig. 5a, the photoconductance ($G_{Ph}$) is plotted versus $V_{BG}$ during

an assay under LED white light illumination (Thorlabs LED diode MCWHF2, cold white). A coherent explanation for the observed photoconductance is that after the light absorption process and generation of an electron-hole pair at the Fc, this pair is broken and electrons contribute to the current under certain $V_{BG}$ bias conditions. Gr carriers are only increased when the channel operates in the electron-conduction regime ($V_{BG} > V_{DP}$), where Gr acts as a better electron acceptor than Fc. In the hole-conduction regime, even if an electron-hole pair is generated, electrons have a very low probability of being injected into Gr. Furthermore, holes cannot be injected into Gr because it is more favorably stabilized as a ferrocenium cation (Fc$^+$) (Fig. 5b, c). Then, bearing in mind a relationship between photoconductance and charge carrier density increment due to light illumination ($G_{Ph} \propto \Delta n_{Ph}$), using Eq. 9 and Eq. 10 in Theoretical Methods, we estimate the number of photo-generated carriers $\Delta n_{Ph} = n_{Light} - n_{Dark} = 3 \times 10^8$. It is in the same order or magnitude as the number of Fc groups on the electrode pore, considering the electrochemical surface density ($n_{molec} = 1.2 \times 10^8$). We completed the study by systematically changing the incident light power and $V_{BG}$ over time. In Fig. 5d the $G_{Ph}$ versus the LED power of the incident light (0.088–8.8 mW) shows a typical Fermi–Dirac distribution with a saturated regime over 5 mWcm$^{-2}$. The same situation is presented for the photoconductance as a function of the $V_{BG}$ (Fig. 5e) with a saturation regime at $V_{BG} = 25$ V (Eq. 11 in Theoretical Methods). Finally, we can observe that there exists an exponential decay (Eq. 12 in Theoretical Methods) on the photocurrent due to the low dynamics of our back gated m-GFET device (Fig. 5f). Good fits are attained with a preexponential factor of 10 C/s and a RC time constant of 80 s. Note that the photoconductance observed here was measured in ambient conditions and the charges trapped in the device, discussed in Fig. 4b, may affect photoresponse[51]. Our m-GFET device can therefore work as a photodetector and may reach high sensitivity thanks to the stability of the ferrocenium that behaves as a trap[12,52] stabilizing the hole while injecting the electron to the Gr at the correct gate bias. Consider that the reported power refers to the diode power output. Considering 0.8 as a typical efficiency for the visible optical fiber and the active surface of 2 pores of 5 μm in diameter, the detected power is roughly 0.7 nW. Additionally, a broad spectral range can be covered depending on the nature of SAMs. Finally, light intensity and $V_{BG}$ can be exploited as inputs for operation of the m-GFET design like a logic gate device. In a prospective way, we reproduce in Fig. 5g the operation of the m-GFET as AND (OR) heterogeneous logic gate. Here the optical gate is operated between 0 and 8.8 (2 and 8.8) mW cm$^{-2}$ for "0" and "1" values, and the electrical back gate is operated at 0 and 50 (10 and 50) V for "0" and "1". The waveforms are set to "10101" and "10011" respectively. As a result, considering a threshold conductance of 6 μS we obtain "10001" ("10111") output, that corresponds with the AND (OR) logical operation.

In summary, we have described the fabrication and operation of a novel hybrid molecular Gr platform. The molecular graphene field-effect transistor (m-GFET) presents potential scalability based on FcC$_{11}$SH SAMs and CVD-grown Gr. First, the Fc group has been useful to follow crucial aspects in the SAM as the $E_H$, $\Gamma$, $\varphi$. Moreover, the properties of Gr such as superior electrical conductivity, remarkable elasticity, membrane impermeability and optical transparency have been key to electron transfer, soft contact, encapsulation and operando spectroscopies. Cyclic Voltammetry shows electron transfer underneath the Gr monolayer and surface spectroscopies indicate that molecules encapsulated by Gr remain in their oxidation state. Therefore, Fc//Gr hybrid SAMs may be considered for long term studies and applications[53]. Additionally, operando XPS and XAS spectra do not show an obvious change in the bias but with the time under synchrotron x-ray illumination. The rectification characteristic is lost by changing the top electrode and DFT calculations support the low rectification ratio of FcC$_{11}$SH with a Gr top electrode. In addition,

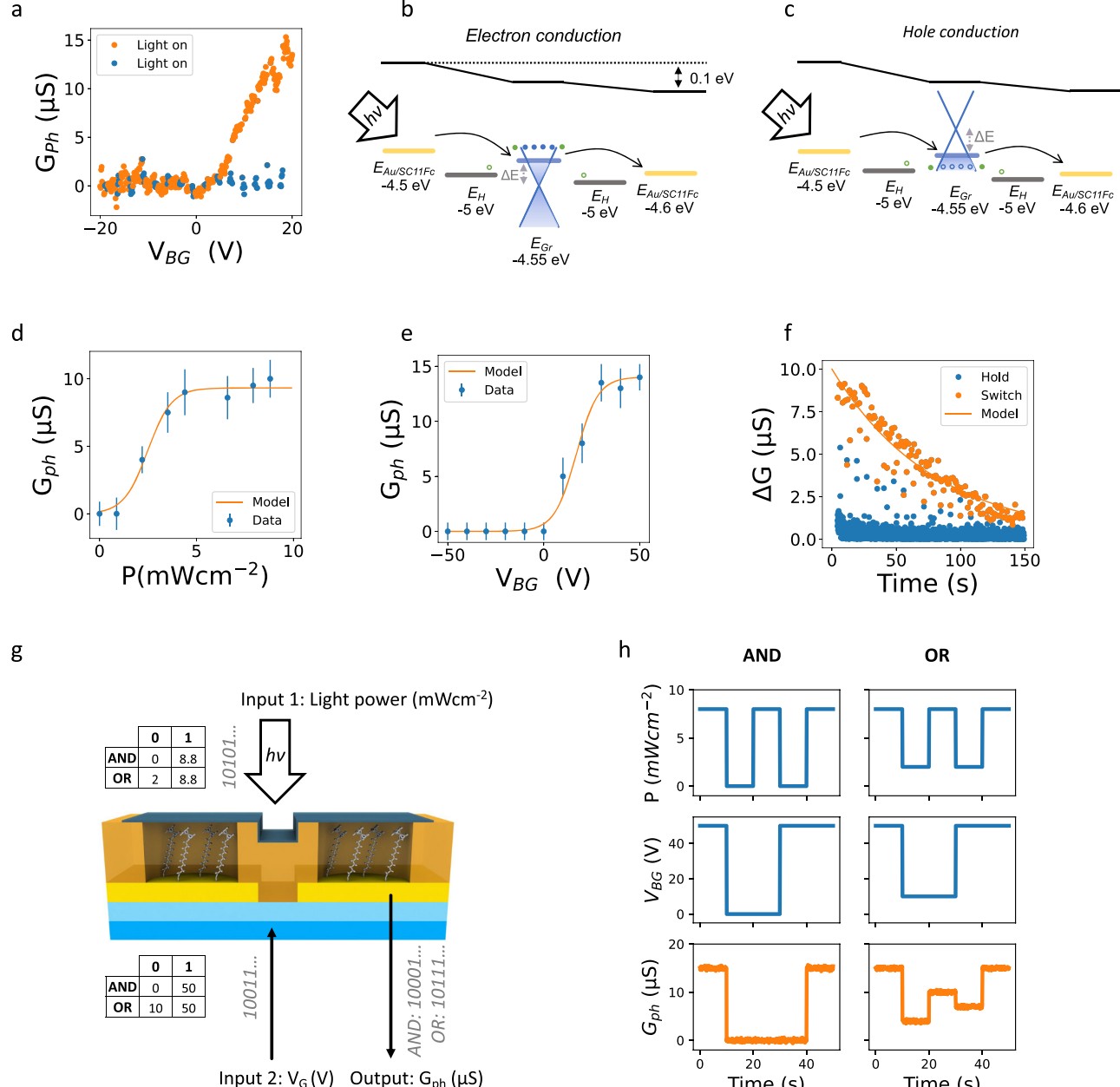

**Fig. 5 | Photo-Electrical properties of the m-GFET. a** Source-drain photo-conductance is displayed versus the gate voltage at ambient conditions during pulsed light illumination (21.5 mW LED). Electrons excited by the light in the Fc group do not affect the photoconducatance if $V_{BG} < V_{DP}$ ($V_{DP}$, is the back gate voltage at the conductance minimum). However, electrons are injected to the Gr conduction band when $V_{BG} > V_{DP}$ giving rise to light photodetection. **b** Schematic illustration of the transport model during light operation for electron and **c** hole conduction. Majority charge carriers, electrons (filled dots) and holes (empty dots) are only increased in de electron conduction regime ($V_{BG} > V_{DP}$). **d** Source to drain conductance under pulsed light illumination at $V_{BG} = 10$ V ($V_{BG} > V_{DP}$) at different light powers 8.8 mW (100%, 75%, 50%, 25 and 0%). **e** Source to drain conductance under pulsed light illumination at 8.8 mW light power versus $V_{BG}$. Error bars represent the standard deviation of the experimental measured $G_{Ph}$. **f** Source to

drain conductance increments under pulsed light illumination at 8.8 mW at $V_{BG} = 10$ V vs the time. $V_{DS} = 10$ mV. **g** Schematic illustration of the heterogeneous dual gate operation. Input 1 comes from white light illumination ($hv$), input 2 is the electrical backgate voltage ($V_{BG}$) the output is the photoconductance ($G_{Ph}$). **h** Reproduced AND operation with 0 and 8.8 mW cm$^{-2}$ and 0 and 50 V for the "0" and "1" values. Input 1 waveform is set to "10101" and input 2 to "10011" resulting in the "10001" output. The OR operation uses 2 and 8.8 mW cm$^{-2}$ and 10 and 50 V for digital values "0" and "1" obtaining "10111". A threshold conductance of 10 μS is set. Power is related to the Power Output of the Thorlabs LED diode MCWHF2. In **d**, **e**, and **f** fits are obtained using Eqs. 12 and 13 and values described in Experimental Methods with Non-Linear Least-Squares Minimization and Curve-Fitting for Python.

molecular decoration tunes the working function of the electrodes and subsequently the Dirac point of the m-GFET device. The ability to back gate the Gr allows passing from poor to good electron acceptor where we can operate the transistor as a photodetector. FcC$_{11}$SH molecules have been integrated into a scalable transistor to make use of their photoelectrochemical properties in a solid-state device.

Additionally, this configuration may produce ultrasensitive photodetectors based on large ferrocenium stability and optical circuits may benefit from this design. Finally, the m-GFET device has been operated as an heterogenous AND/OR logic function. In future studies we will explore the electromechanics and biocompatibility of these m-GFET devices.

## Methods

### Experimental design

We first explore the use of graphene as a high-performance top contact electrode on a single junction showing that: (1) electron transfer between the Fc group in the Au/SC$_{11}$Fc SAM and the gold electrode is achieved by applying an electrochemical voltage and consequently generating the electrochemical double layer through a Gr monolayer that adjusts the energy level of the system; (2) surface spectroscopy shows that Gr encapsulates the molecules avoiding ambient Fc oxidation and (3) operando examination helps to discuss the transport mechanism in the molecular junction. On the m-GFET, (4) molecular decoration of electrodes shifts the graphene V$_{DP}$ to neutrality and (5) photodetection based on Au/SC11Fc //Gr electron transfer is realized on the electron conduction regime. Finally, (6) the heterogenous optical and electrical dual gate is operated as OR/AND logic functions. The archetype 11-(Ferrocenyl)undecanethiol (FcC$_{11}$SH) is acquired from Sigmaaldrich and characterized with electrochemical and spectroscopic methods (XPS and XAS) once assembled on the surface. This redox couple allows to estimate the quality of the SAM by a surface coverage and introduce a redox function to access by the light when the molecule is mounted in the m-GFET. μSAMs are formed on the gold surfaces not covered by the HfO$_2$.

### Device fabrication

W-Oxide Wafer 04/009 100 mm. N PRIME CZ 100As 0.001–0.005 ohm.cm 525+/−20um Single side polished Semi standard flat Oxide−AtOx, 290 nm Double side coated. Electrodes: The device consists of source and drain chromium/gold micrometer size electrodes. The distance between drain and source is 20 μm and the electrodes sit on top of a SiO$_2$/Si substrate (290 nm of SiO$_2$ and highly n$^{++}$ doped silicon). The electrodes are passivated by a layer of hafnium oxide (HfO$_2$) and the molecular junctions integrated in a micropore, both electrodes are contacted together thanks to a CVD graphene that is back gated. The micropores on the electrodes are etched by HF (1%) treatment during 5 min. μSAMs are formed by immersing the surfaces in a freshly ethanolic solution of 3 mM of the ferrocenyl undecanethiol for 12 h. The samples are then taken out from the solution, rinsed with ethanol, and dried under N$_2$ stream. CVD graphene Transfer: Homemade and commercially available from Graphenea, high-quality monolayer graphene (SLG) grown by CVD technique on copper foil (Cu), is transferred by standard PMMA transfer process onto $1 \times 1$ cm$^2$ SiO$_2$/Si substrates where the final devices are fabricated as explained above. First, a sacrificial PMMA layer (495, A4; 5000 rpm, 60 s; baked for 1 min at 180 °C) is spin coated on SLG/Cu and the graphene on the Cu backside etched by oxygen plasma (100 W; 2 min; 150 ml/min O$_2$). Then, the Cu layer is wet etched using an ammonium persulfate solution (APS), obtaining a PMMA/SLG film. The PMMA/SLG stack picked up from solution, using a glass slide, to a DI water bath to remove chemical residuals. After three DI water baths (10 min each), the PMMA/SLG film is transferred on the final device with the corresponding molecular SAMs. Then, the stacked PMMA/SLG /substrate is dried in air for 30 min and stored in vacuum for at least 24 h, to avoid the detachment of the graphene from the substrate. Finally, the PMMA layer is removed dipping the sample in acetone for 1 h and in isopropyl alcohol for another hour.

### Electrochemical measurements

Experiments are performed in a three-electrode, single chamber Teflon cell connected to an Autolab PGSTAT204 (Metrohm) Electrochemical Analyzer in 0.1 M NaClO$_4$ electrolyte using an Ag/AgCl reference electrode and a Pt wire as a counter electrode. Before the experimental measurements, the electrochemical cell was cleaned with ethanol and DI water. 'Test' sweeps between 0.1 and 0.6 V with a highly doped silicon substrate are measured to confirm that there is no peak due to contamination.

### AFM and CAFM

Experiments are conducted in a Nanotec Cervantes AFM (Nanotec). ElectriMulti75-G AFM probes (Budget Sensors) are used with platinum overall coating and a nominal force of spring constant 3 N/m and nominal resonance frequency of 75 kHz. All AFM raw data are analyzed with WSxM software[54].

### XPS

Experimental characterization is produced using Ultra-High Vacuum (UHV) conditions with a photon energy of 1486.7 eV (Al Kα transition).

### UPS

Experimental characterization is produced using UHV conditions with a He I radiation of a helium gas discharge lamp Specs 10/35 model and a hemispherical energy analyzer (SPHERA-U7).

### Spatially resolved X-ray absorption spectroscopy (XAS)

Experimental characterization was performed at the CIRCE beamline of the ALBA synchrotron Light Facility. The PEEM endstation is equipped with an Elmitec LEEM/PEEM III microscope. The microscope can be used as a regular LEEM microscope using electrons as illumination source or as a Photo-Emission Electron Microscope by illuminating the sample with X-rays. The spatial resolution is 20 nm and the beamline provides photon energies in the soft X-ray range, from 100 eV to 2000 eV with high intensity and resolution, and fully variable polarization. Low-energy secondary electrons are used to form images of the spatially dependent X-ray Absorption Spectrum.

### Raman spectra

Experiments are acquired with a Bruker Senterra confocal Raman microscope instrument equipped with a 532 nm excitation laser. Acquisition parameters are: 2 mW, 5 s integration time, 10 co-additions, 1–15 cm$^{-1}$ resolution.

### Optoelectrical measurements

Experiments was carried out on a Keithley 4200-SCS probe station with four SMU using an incident light from a diode, Thorlabs Fiber-Coupled (LED) MCWHF2.

### Electrochemical current

It is modeled by the extended Laviron model[22]:

$$i(\theta_0) = \frac{n^2 F^2 A \upsilon \Gamma_{max}}{RT} \frac{\theta_0 (\theta - \theta_0) \theta}{\theta^2 - 2 G \theta_0 \phi(\theta)(\theta - \theta_0)} \tag{1}$$

Where $i(\theta_O)$ is the measured voltammetry current, $A$ is the electrochemical working electrode area, $\upsilon$ is the potential scan rate (Vs$^{-1}$), $\Gamma_{max}$ is the surface coverage (mol cm$^{-2}$), $n$ is the number of electrons exchange per molecule during a redox process, $F$ is the Faraday constant, $R$ is the Universal Gas constant, $T$ is the temperature, $\theta_O$ the oxidized normalized surface coverage, $\phi(\theta)$ a segregation factor, and $G$ is a global constant of interaction.

### Molecular energy level

Molecular energy level in the junction is extracted from the electrochemical measurements[26]:

$$E_{HOMO} = E_{abs,NHE} - e E_{\frac{1}{2},NHE} \tag{2}$$

$E_{abs,\ NHE}$: absolute potential energy of the normal hydrogen electrode −4.5 eV.

$e$: elementary charge, $1.602 \times 10^{-19}$ C.

$E_{1/2,\ NHE}$: formal half-wave potential versus normal hydrogen electrode.

Ag/AgCl as reference electrode vs. NHE. 0.197 V

## Surface coverage

The $\Gamma_{Fc}$ (mol/cm²) is calculated by Eq. 3, where the $Q_{Fc}$ is the total charge calculated by the integration of the redox waves, $n$ is the number of electrons interchanged per molecule in the reaction, $F$ is the Faraday constant and $A$ the surface of the working electrode exposed to the electrolyte solution (0.2 cm²)

$$\Gamma_{Fc} = Q_{Fc}/nFA \qquad (3)$$

## Coulombic interactions

They can be estimated from a simple model[12]:

$$\varphi = e \frac{1 - \left[1 + \left(\frac{r_a}{d}\right)^2\right]^{-0.5}}{4\pi\varepsilon_0\varepsilon_r d} \qquad (4)$$

Considering $e$ the elementary charge, $r_a$ the ion pairing distance, $d$ intermolecular distance, $\varepsilon_O$ the dielectric permittivity of vacuum and $\varepsilon_r$ the relative permittivity of water respectively and $\varphi$ an average value per molecule-molecule coulombic interaction.

## Energy molecular junction alignment

$$\Phi_{Au} > E_H + V \, \eta > \Phi_{Gr} + V \qquad (5)$$

$$\Phi_{Au} < E_H + V \, \eta < \Phi_{Gr} + V \qquad (6)$$

$\Phi_{Au}$: Gold work function, $\Phi_{Gr}$: Graphene work function, $E_H$: Molecular energy level,
$V$: Applied voltage, $\eta$: voltage drop

## DFT calculations

As a support to interpret the experimental results, we have modeled the junction to determine theoretically its electronic properties and its electronic transport characteristics, in particular its response to an applied bias. To this end, we have considered a model junction (as shown in Fig. 4a), namely a FcC₁₁SH molecule sandwiched between a 35 gold atom pyramid acting as a gold electrode, and a 5 × 5 unit cell of graphene, which is repeated periodically in the $xy$ plane, mimicking an infinite graphene plane, to act as the second electrode. This structure has been fully optimized, until the forces went below 0.1 eV/Å. We have used the very efficient localized-orbitals basis set DFT code Fireball[55]. Basis sets of $sp_3d_5$ numerical orbitals for Au and Fe, $sp_3$ for C and S, and s for H have been used for structural optimization and conductance calculation of the nanojunctions, with cutoff radii (in atomic units) $s = 4.5$, $p = 4.9$, $d = 4.3$ (Au), $s = 4.8$, $p = 5.0$, $d = 5.2$ (Fe), $s = 4.5$, $p = 4.5$ (C), $s = 3.1$, $p = 3.9$ (S), and $s = 4.1$ (H)[56]. A formalism taking into account van der Waals interactions has also been considered to determine the molecule–graphene distance[57]. From the optimized configuration, we can determine the electronic structure of the system, and use it to determine the electronic conductance and the electronic current in the molecular junction. We use here a non-equilibrium Keldysh-Green formalism, which has been designed initially to calculate simulated STM images, but which can be used to calculate electronic current in a molecular junction since it takes into account multiple scattering[58]. This formalism has been already applied successfully to several types of molecular junctions[35,59].

## Dirac point shift

Photoconductance versus time follows this model as the electric charge.

$$\triangle E = \hbar v_F \sqrt{\pi n \left(V_{D2} - V_{D1}\right)} \qquad (7)$$

$\hbar$: Planck constant in J s rad⁻¹, $v_F$: Fermi level velocity $1.1 \times 10^6$ m s⁻¹, $\pi$: Mathematical constant, $n$: carrier density $7.2 \times 10^{10}$ cm⁻² V⁻¹, $V_{D2}$: Change of neutrality point with respect the reference.

## Trapped charges

The number of charges can be calculated with the following expression:

$$N = C_{ox}\triangle V/2e \qquad (8)$$

$C_{ox} = 115$ µFm⁻², $\Delta V = 60$ V, $e = 1.6 \; 10^{-19}$ C.

## Graphene mobility

*Graphene mobility ($\mu$, cm²V⁻¹s⁻¹) is calculated with the following geometrical characteristics. $L = 10$ µm (graphene length), $W = 24$ µm (graphene width), $C_{ox} = 115$ µFm⁻², $V_{DS}$: Drain to source voltage, $I_{DS}$: Drain to source current, $V_{BG}$: Back gate voltage, $g_m$: transconductance.*

$$\mu = \frac{L}{W}\frac{1}{C_{ox}V_{DS}}g_m \qquad (9)$$

$$g_m = \frac{\partial I_{DS}}{\partial V_G}$$

## Photoconductance

$$\frac{G_{Ph}}{G_{Dark}} \approx \frac{\triangle n}{n_{Dark}} \qquad (10)$$

$G_{Ph} = 10$ µS (photoconductance), $G_{Dark} = 2$ mS (conductance at dark state), $n_{Dark} = 7.2 \times 10^{10}$ cm⁻²V⁻¹ (dark state number of carriers), $\Delta n = n_{Ligh} - n_{Dark} = 3 \times 10^8$ (increment in the number of carriers).

From the linear dependence of conductivity, $\sigma$ ($V_{BG}$), elementary charge, $e$, carrier mobility, $\mu$, we can find the number of carriers, $n$[60].

$$\sigma = \mu \, n \, e \qquad (11)$$

## Fermi−Dirac distribution

Photoconductance versus power and gate voltages is modeled by the logistic model. It determines the statistical distribution of fermions over the energy states of a system in thermal equilibrium.

$$\bar{n}_i = \frac{1}{e^{\frac{(\varepsilon_i - \mu)}{k_b T}} + 1} = \frac{A}{1 + e^{\frac{4\mu}{A}(\lambda - t) + 2}} \qquad (12)$$

where $k_B$ is Boltzmann's constant, T is the absolute temperature, $\varepsilon_i$ is the energy of the single-particle state i, and $\mu$ is the total chemical potential. Second term in the mathematical expression is a reparameterization logistic model used to fit the experimental data. $A = 9.32$, $\mu = 3.78$, $\lambda = 1.25$ for fits in Fig. 5d and $A = 13.95$, $\mu = 0.54$, $\lambda = 3.27$ for fits in Fig. 5e.

## Exponential decay model

Photoconductance versus time follows this model as the electric charge. $V_0/R = 10$ C/s and RC = 80 s for fits in Fig. 5d.

$$I = \frac{V_0}{R} e^{-t/RC} \qquad (13)$$

## Data availability

The datasets generated during and/or analysed during the current study are available from the corresponding author on reasonable

request and at IMDEA repository https://repositorio.imdeananociencia.org/handle/20.500.12614/3261.

## Code availability

The code that supports the findings of this study is available from the corresponding author upon request.

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

## Acknowledgements

We acknowledge the Comunidad de Madrid projects NMAT-2D-P2018/NMT-451 and TEC2SPACE-CM P2018/NMT-4291. IMDEA Nanociencia thanks support from the "Severo Ochoa" Programme for Centers of Excellence in R&D (MINECO, Grant CEX2020-001039-S). This work was also supported by Ministry of Science, Innovation and Universities under Grants PID2019-105552RB-C4-1 and ESP2017-92706-EXP, MADE-MICINN: PID2019-105552RB-C44. We also acknowledge financial support from ONR-Global under Grant DEFROST N62909-19-1-2053 and EMPIR-20FUN06-MEMQuD-EURAMET. MAN, MF and MWK acknowledge funding by MICINN through PID 2021-122980OB-C54. MWK acknowledges funding from Horizon 2020 Marie Skłodowska-Curie COFUND DOC-FAM, with Grant agreement No. 754397. PEEM experiments were performed at CIRCE beamline at ALBA Synchrotron Light Facility with experiment ID AV-2020024142. We thank comments and suggestions of Dominique Vuillaume and the work of David Cantarero Tomas for the production of the artistic Fig. 3, info@davidcantarerotomas.com.

## Author contributions

J.T. and D.G. conceived the general research. J.T. designed the experiments, fabricated the devices, carried out the electrochemical, electrical and optoelectrical experiments, analysed the data, and proposed mechanisms. MAN and JCM-R carried out the XPS experiments and analyzed the data. J.T., M.W.K., S.R.-G., M.F., and M.A.N. obtained and analyzed the XAS data. M.C.d.O. get done Raman experiments and analysed the data. Y.D. performed the DFT calculations. J.T., J.G.-P., M.A., M.R.O., M.T.M., M.C.d.O., A.G. and D.G. supported the fabrication of devices. J.T. and P.P. performed the A.F.M. and CAFM experiments. R.M. and D.G. managed the research project. Results were discussed by all the authors. The manuscript was written by J.T. with improvements of D.G. and contributions from all the authors.

## Competing interests

The authors declare no competing interests.
