## [Peer Review File · Nature Communications]

Hybrid molecular graphene transistor as an operando and optoelectronic platformREVIEWER COMMENTS

Reviewer #1 (Remarks to the Author):

The authors report a hybrid molecular graphene field effect transistor made with 11-(Ferrocenyl)undecanethiol and high-quality graphene (Gr) in a back-gated field effect transistor configuration. Dual gating operation by light illumination and gate voltage is verified. Such a scalable m-GFET device has been operated as an heterogenous AND/OR logic function. The work of this paper is well-written and solid. However, there are some problems that should be further improved as follows:

Remarks:

- 1. In line 22, the poor reproducibility in measurements should be introduced more specifically and give possible reasons. The reproducibility in measurements and success rate for device fabrication in this research should be specific in the article.**
- 2. In supplementary Note 5, it is mentioned that the Dirac point is close to the Fermi level, which is attributed to the strong interaction on the other side between gold and thiol. How can this speculation be proved experimentally?**
- 3. In line 79, two different transport mechanisms in reverse vs forward polarizations are mentioned. Are there the temperature-dependent measurement results?**
- 4. In part Operando charge transport in the ferrocenyl m-GFETs, the authors should add how to measure and quantify the trapped charges in the device, and how to avoid the trapped charges?**
- 5. In line 194, the oxidation state of Fc can't be changed by the applied bias. Can the oxidation be achieved in other field effect transistors, and how did the authors oxidize the Fc in a molecular graphene FET?**
- 6. In figure 4d, the absorption is different at different source to drain voltages. What is the reason for this?**
- 7. Figure 1e and Figure S23 could be clearer, please check them.**

Reviewer #2 (Remarks to the Author):

The authors studied molecular graphene field-effect transistors (m-GFETs) made from ferrocene molecules (FcC11SH). They performed a series of measurements on various junction structures such as Au/SC11Fc, Au/SC11Fc//Gr, Au/SC11Fc//EGaIn, GEET, and m-GFET. My general impression of this study is that it was rather distracting as it presented various junction structures. It is like a study that combines a series of independent studies. Authors should focus more on the point, and the point should be on the m-GFET. And while the results in Figure 5 are interesting, but the results are not that meaningful because of using two different inputs: light and electrical signals. Considering all these, I am not willing to recommend this manuscript to Nature Communications. The authors should focus more on the point and then submit elsewhere. I have following comments for authors to revise the manuscript.

- (1) Line 114. Regarding "Especially, for large voltage windows, there is a forward regime at positive voltages that should cause a regime with negative differential resistance", authors should present experimental data or theoretical data with NDR.**
- (2). Line 115. Regarding "Considering this approach, the lack of rectification in covered junctions may derive from the fact that Gr affects the voltage drop promoting to work in the no rectification regime (close to the black circle)", I can't find black circle in Figure 2c.**
- (3) I can't find any paragraph that explains the results of Figure 4e and 4f in the manuscript.**
- (4) The results of Figure 5 are not that meaningful because of using two different inputs (light and electrical signals). I would be rather more convinced if authors demonstrate**

logic with using only electric signals.

Reviewer #3 (Remarks to the Author):

Trasobares et al. designed a hybrid molecular graphene FET with Au/FcC11SH as the source and drain electrode. By tuning the carrier transport between FcC11SH and graphene, the Dirac point of graphene in transistors can be modulated and the device can respond to both optical and electrical input. The device structure is new and the results are quite interesting. The use of graphene and ferrocene molecules introduces more functions to the device. However, the presentation and the discussion of the results are poor. The authors should focus on the most important breakthrough and show the novelty of this work clearly. The manuscript should be revised thoroughly to meet the high standard of Nature Communications.

1. In the title, the abstract, and the introduction, the authors emphasized the importance of reproducibility in molecular electronics and the lack of operando characterization. If the authors believed that this is the most striking part of the results, more experimental and theoretical results should be displayed, not just the XAS spectra in Figure 4d.

2. In fact, several figures contain the band diagrams (Fig 2ab, 4ef, 5bc) and the authors discussed the charge transport at the molecule//graphene interface. More evidence should be provided to demonstrate these mechanisms. An in-depth understanding of the unique electronic and optoelectronic transport at the interface is helpful to design molecular electronics. Chem, 2020, 6(5), 1172-1182 gives some results in this aspect. They observed a low-conductance state in the oxidized state of ferrocene with symmetrical characteristics, while a high-conductance state in the reduced state with asymmetrical curves.

3. Some descriptions are not correct. For example, lines 181-182, "Each VBG range generates different number of charge trapped at the SiO₂//Gr interface" The interface here should be HfO₂//Gr.

4. In figure 4, the authors talked about the charge traps in the device and showed the hysteresis results. However, this is not the intrinsic behavior of molecular electronic devices. The fabrication process should be optimized to minimize this effect, such as the deposition of a high-quality HfO₂ layer or the annealing. It is better to focus on the main breakthrough of this work. Some irrelevant discussions can be moved to SI.

5. The device can respond to optical stimuli, which is quite interesting. The authors believe that light is absorbed by the molecules. The absorption spectra are absent. And a control experiment without molecules can also demonstrate it because graphene can absorb 2.3% light per layer in a broadband spectrum.

REVIEWER COMMENTS

Reviewer #1 (Remarks to the Author):

The authors report a hybrid molecular graphene field effect transistor made with 11-(Ferrocenyl)undecanethiol and high-quality graphene (Gr) in a back-gated field effect transistor configuration. Dual gating operation by light illumination and gate voltage is verified. Such a scalable m-GFET device has been operated as an heterogenous AND/OR logic function. The work of this paper is well-written and solid. However, there are some problems that should be further improved as follows:

We sincerely acknowledge the reviewer for the careful review and suggestions. Below, we answer point by point to the comments.

Remarks:

1. In line 22, the poor reproducibility in measurements should be introduced more specifically and give possible reasons. The reproducibility in measurements and success rate for device fabrication in this research should be specific in the article.

We thank the referee for the constructive comments. This is a key point that we have now extended in the manuscript with the following changes including additional results.

In line 23:

“..., differences in the resistance per molecule measured through molecular junctions can be up to 8 orders of magnitude, this feature is often attributed to variances in contacts, surface roughness, presence of defects and how all these factors scale with the junction size⁶.”

Line 161:

Using this protocol, we have an operating yield of 65%.

Note 6:

“We have used 3 wafers (4 inch) containing 80 dies with 9 devices each die. Overall, 70.6% of devices results as the one in Figure S12b. 17.1% presented some lithographic defects either on the electrodes or resin residues and 12.3% of the devices failed during the graphene transfer process.”

Note 7:

“Regardless the appearance under the optical microscope. The electrical operation of the device is validated via the source to drain current versus the gate voltage. The position of the Dirac point is used to confirm their proper operation. Considering that the molecular functionalization of the device tunes the V_{DP} to neutrality by changing the working function of the electrode, we assume working devices those with a V_{DP} close to neutrality.

Alternatively, samples with a $V_{DP} > 30$ V are assumed to be defective. As we presented in Figure S21, 92% of the fabricated device shift the V_{DP} to neutrality”

Figure S21. Dirac point position, G_{DS} vs. V_{BG} of the 500 m-GFETs.

2. In supplementary Note 5, it is mentioned that the Dirac point is close to the Fermi level, which is attributed to the strong interaction on the other side between gold and thiol. How can this speculation be proved experimentally?

We agree with the Referee that this paragraph has been wrongly formulated and that the strong interaction between gold and the molecule to explain the neutrality of graphene may sound speculative. We have rewritten the paragraph in the following way:

“We can notice from these calculations that the Dirac point (found in Γ due to the 3x3 unit cell) is very close to the Fermi level. That indicates there is no doping of graphene by ferrocene. Additionally, the total charge of graphene on Fc SAMs is almost equivalent to that of isolated graphene. This is easily explained by the π - π interactions between ferrocene and graphene that minimize energy level broadening¹⁶.

In this work we have exploited operando spectroscopic techniques such as: XPS and XAS determining the state of the main atoms (Fe, S, C) involved in the device. Additionally we performed: UPS, getting values of the work function of the device and threshold photoemission electron microscopy (th-PEEM), obtaining topographic, work function and chemical contrast (now included in Supplementary Information, Note 9. Ultraviolet photoelectron spectroscopy, X-ray absorption spectroscopy, threshold Photoemission electron microscopy).

3. In line 79, two different transport mechanisms in reverse vs forward polarizations are mentioned. Are there the temperature-dependent measurement results?

We thank the referee for the relevant remark. We have restructured the sentence to clarify that our in-operando approach may be used as an alternative to temperature dependent measurements established in the mentioned references.

“As first proposed by C. A. Nijhuis et al, the rectification in these 11-(ferrocenyl)-1-undecanethiol (SC11Fc) junctions²⁵⁻²⁷ emerges from two different transport mechanisms in reverse (bond tunneling) vs forward (bond tunneling and sequential tunneling) polarization^{13,15,26}. Usually, temperature-dependent measurements are applied to distinguish those regimes²⁷. Our graphene covering 11-(ferrocenyl)-1-undecanethiol (SC11Fc) hybrid SAMs can overpass this tedious temperature dependent characterization or limitations by enabling operando spectroscopy. For example, our approach may present advantages in junctions pushed to work in the inverted Marcus Regime where the transport mechanism becomes activationless²⁸”

4. In part Operando charge transport in the ferrocenyl m-GFETs, the authors should add how to measure and quantify the trapped charges in the device, and how to avoid the trapped charges?

We thank the referee for this pertinent observation. Trapped charges, commonly related with impurities or defects disturb the Dirac point. Some researchers⁴³ have elaborated a self-consistent theory to explain the existence, position and width of the minimum conductivity point. For example, this minimum conductivity may vary from $\sigma_0 \approx 4e^2/h$ to $\sigma_0 \approx 8e^2/h$ for dirty and clean samples respectively. Others⁴⁴, have related the position of the neutrality point to charge transfer and capacitive gating. Following the latest method, in a simple case where the shift is related with trapped charges, the number of charges can be evaluated with the following expression: $N = C_{ox}\Delta V/2e$. For instance, in our devices, a positive shift of 56 V when sweeping $\pm 100V$ relates with a trapped charge density of $2 \times 10^{12} \text{ cm}^{-2}$ ($C_{ox} = 115 \mu\text{Fm}^{-2}$).

In order to avoid this hysteresis, the conventional approach is the encapsulation by Al_2O_3 under proper process conditions. It enables hysteresis free and reproducible device characteristics indicating that the trapping sites are reduced⁴⁴. Note that thermally grown silicon oxide has effective trap densities $N \approx 10^{10}$ and 10^{11} cm^{-2} for interface and oxide traps, respectively⁴³. Then, a more interesting alternative might be a novel approach consisting in tuning the Fermi level of the channel to maximize the energy distance between the charge carriers in the channel and the defect bands in the amorphous aluminium gate oxide⁴⁵.

We introduce the equation 8 in the Methods section:

“Trapped charges: The number of charges can be calculated with the following expression: $N = C_{ox}\Delta V/2e$ (8)

$C_{ox} = 115 \mu\text{Fm}^{-2}$, $\Delta V = 60 \text{ V}$, $e = 1.6 \cdot 10^{-19}\text{C}$.”

We introduce the Table 2 in the Note 7 of the Supplementary Information:

Table S2. Number of charged trapped.

V_{BG} (V)	V_{DP-} (V)	V_{DP+} (V)	Shift DP (V)	$N(\text{cm}^{-2})$
10	-1.4	0.22	0.81	2.91E+10
50	-17	14.13	15.565	5.59E+11
75	-32	36	34	1.22E+12
100	-51.7	60	55.85	2.00E+12
150	-95	105	100	3.59E+12

In Line 210:

~~By increasing the V_{BG} range, we observe tuning of the Dirac point up to 120 V (Figure 4b). Each V_{BG} range generates different number of charge trapped at the SiO_2/Gr interface and subsequently diverse false doped states that shift the Dirac point.~~

“However, when we increase the V_{GB} the hysteresis increases, this is an indication that charges are injected into the interface or bulk SiO_2 ^{43,44}. Number of trapped charges (N) can be calculated with Equation 8 in Methods. When the V_{BG} is swept $\pm 100 \text{ V}$ there is a Dirac point shift of 55 V, this is equivalent to a trapped charge density of $2 \times 10^{12} \text{cm}^{-2}$ (calculated N for the ranges plotted in Figure 4b are shown in Table S2). Although in many applications trapped charges may be undesirables^{45,46}, this hysteretic mechanism may be rich for the purpose of nonvolatile memory devices and give extra performances in chemical or biological sensing.”

5. In line 194, the oxidation state of Fc can't be changed by the applied bias. Can the oxidation be achieved in other field effect transistors, and how did the authors oxidize the Fc in a molecular graphene FET?

We thank the comment. Some authors as in references 21 and 33 assume that the separation between the top electrode (graphene) and the Fc-group may result from their mutual electrostatic repulsion upon oxidation. In our devices we do not see a change in the conductivity (rectification) neither a change in the oxidation state (XAS). We think that the mechanical forces may affect geometry and are subject of current investigations. However, we note that the photodetection observed in the graphene electron conductivity regime may be caused by the photoinduced oxidation of ferrocene by incident light.

We modify part of the paragraph (Line 235):

“Laterally resolved X-ray absorption spectroscopy (XAS) performed at the ALBA synchrotron radiation facility with a PEEM microscope shows in the electrode pore a clear Fe signal at the Fe L3 edge during all the experiment (Supplementary Note 9 and Figure 4d). XAS spectra confirm the grafting of the ferrocenyl μ SAMs only at the Au exposed pore under Gr. In opposite to our results in Figure 1c and other configurations where authors demonstrate chemical and electrochemical ferrocene oxidation and reduction by applying a voltage to an electrolyte on top of the graphene²¹, the oxidation state of Fc is independent of the source-drain voltage remaining in the reduction state.”

6. In figure 4d, the absorption is different at different source to drain voltages. What is the reason for this?

We thank the referee for this comment. We have observed a change on the Fe absorption edge for different drain voltages, changes towards an increased oxidation state but this photochemical switching is mostly due, as we mention on the text of the paper, to the intense photon irradiation (we have updated the legend of Figure 4d for clarity). Some organic molecules suffer changes in the electronic state, molecular bondings, etc. under intense photon irradiation, changes that in laboratory sources are less important but in synchrotron radiation sources are more important due to the high photon flux that increase the probability of changes. Although we do not rule out possible differences due to different drain voltages, we believe the main changes are due to the photon irradiation, as it has been reported in Ref. 48. Although, there is an important difference in our experiment respect the reference work that the graphene layer is changing the evolution from a reduced state (in a direct exposed ferrocene) to an oxide state (in a graphene capped ferrocene). We have now updated the legend in Figure 4d, including the time elapsed at the end of the measure.

7. Figure 1e and Figure S23 could be clearer, please check them.

Thank you very much for this information. We corrected this error caused when converting word to pdf.

Reviewer #2 (Remarks to the Author):

The authors studied molecular graphene field-effect transistors (m-GFETs) made from ferrocene molecules (FcC₁₁SH). They performed a series of measurements on various junction structures such as Au/SC₁₁Fc, Au/SC₁₁Fc//Gr, Au/SC₁₁Fc//EGaIn, GEET, and m-GFET. My general impression of this study is that it was rather distracting as it presented various junction structures. It is like a study that combines a series of independent studies. Authors should focus more on the point, and the point should be on the m-GFET. And while the results in Figure 5 are interesting, but the results are not that meaningful because of using two different inputs: light and electrical signals. Considering all these, I am not willing to recommend this manuscript to Nature Communications. The authors should focus more on the point and then submit elsewhere. I have following comments for authors to revise the manuscript.

We would like to thank very much the referee for their work and comments. We are sorry that the study is not of their interest and hope that with these answers the manuscript will be improved.

We emphasize that the focus of the work is indeed the innovation of a novel platform that to the best of our knowledge has never been built. This platform allows access to light for operando characterization and/or manipulation as an extra gate. The first part of the work focuses on the characterization of the hybrid material ferrocene//graphene and the second part is the integration of the same hybrid material in the development of an electrical transistor. The research and the paper are conceived and written as a single work, but it arrives gradually. This transistor holds photosensitive properties that among other functions can bridge optical and electrical circuits, or employed to sense the electromagnetic spectrum in a very accurate way, for example, by varying the active molecule (in this case ferrocene) for each spectrum range.

The significance of the results shown in Figure 5 is to make the reader understand that our new platform extends the properties of a graphene-based field-effect transistor. Although this concept is further extended in answer 4, a simple GFET (just like our m-GFET) can perform logic functions only with electrical sources. However, between these two devices only our m-GFET could combine optical and electrical signals.

In this regard, we modify:

Line 42:

In this work, we first investigate the properties of a single SAM//Gr hybrid junction and then introduce the same material in a new optoelectronic platform (m-GFET) based on Gr and micro self-assembled monolayers (μ SAMs) made with FcC₁₁SH. Our novel design, free of ionic liquid, uses Gr as a soft, high conductive and transparent top contact that encapsulates the molecular entity of interest, allowing both operando characterization and optical gating.

Line 145:

“With all the previous information we now implement this Fc//Gr hybrid material in a field effect transistor configuration.”

(1) Line 114. Regarding “Especially, for large voltage windows, there is a forward regime at positive voltages that should cause a regime with negative differential resistance”, authors should present experimental data or theoretical data with NDR.

We thank very much the referee’s comment. This sentence aims to point out the relevant work of C. A. Nijhuis and co-workers in reference 34 where they not only extend the rectification ratio due to electrostatic forces between the molecule and the top electrode but also the breakdown voltage. Following the same reasoning used by C. A. Nijhuis in reference 25-27, in our figure 2c, applied voltage versus voltage drop (η), we extend the energy level diagram usually taken with a $\eta = 0.7$. Blue backgrounds represent reverse regimes, when the Energy of the HOMO level does not overlap with the Fermi levels of the electrodes, highlighting the diode reverse zones. Due to extra UPS experiments we have determined the working function of the ferrocenyl decorated electrode and updated our Figure 2c. That makes even more complicated and pointless discussing about NDR.

Then we modify the sentence as:

“Especially, for large voltage windows, unless a carefully consideration³⁴, it may appear a forward producing the breakdown before reaching 2 V.”

(2). Line 115. Regarding “Considering this approach, the lack of rectification in covered junctions may derive from the fact that Gr affects the voltage drop promoting to work in the no rectification regime (close to the black circle)”, I can’t find black circle in Figure 2c.

We appreciate the comment and we are sorry for this error during the format conversion that has made referees' work more difficult. We have now updated our Figure 2c based in new experimental results.

(3) I can’t find any paragraph that explains the results of Figure 4e and 4f in the manuscript.

We thank the comment and now we add a clarification at the end of section: Operando charge transport in the ferrocenyl m-GFETs.

Line 224:

“In view of the above, the usual operation of these transistors is performed with a low voltage of 0.1 V. Where according to the band diagram for electron and hole conduction regimes (found in Figures 4e and 4f) show that the molecular energy level would not come into play. Then, electron transport would be tunneling through the entire molecule.”

(4) The results of Figure 5 are not that meaningful because of using two different inputs (light and electrical signals). I would be rather more convinced if authors demonstrate logic with using only electric signals.

We really appreciate this comment. We also believe it may come from the fact that we have failed to highlight something really important in the paper which is the increase in performance of m-GFETs versus GFETs. In the case where the referee would be more interested with the demonstration of a logic function with only electrical signals, this can be achieved with both GFETs and mG-FETs. We select results from a m-GFET curves due to the well centered Dirac point close to $V_{BG} \approx 0V$. In the figure accompanying this answer there are two curves of I_{SD} vs. V_{BG} for 10 mV and 12mV source to drain voltages.

Figure S22. Electrical signals of a m-GFET with $V_{SD}=10mV$ and $V_{SD}=12mV$.

If we consider input 1 the V_{BG} with a value 0V for a digital value 0 and 10V for a digital value of “1” and input 2 with a magnitude of 10 mV for a digital value of “0” and 12mV for a “1”. We can obtain the following table:

Input 1: V_{BG}	V_{BG} (V)	Input 2: V_{SD}	V_{SD} (mV)	Output: I_{SD} (μA)
0	0	0	10	37
1	10	0	10	45
0	0	1	12	52
1	10	1	12	62

Therefore, if we select a threshold around 55 μA (higher than 52 μA and lower than 62 μA) we can have a “and” function or choose a threshold close to 40 μA (higher than 37 μA and lower than 45 μA) we will get an “or” function utilizing only electrical inputs. This supplementary information is added to Note 7.

Reviewer #3 (Remarks to the Author):

Trasobares et al. designed a hybrid molecular graphene FET with Au/FcC₁₁SH as the source and drain electrode. By tuning the carrier transport between FcC₁₁SH and graphene, the Dirac point of graphene in transistors can be modulated and the device can respond to both optical and electrical input. The device structure is new and the results are quite interesting. The use of graphene and ferrocene molecules introduces more functions to the device. However, the presentation and the discussion of the results are poor. The authors should focus on the most important breakthrough and show the novelty of this work clearly. The manuscript should be revised thoroughly to meet the high standard of Nature Communications.

We greatly thank the referee for the very relevant and constructive suggestions.

We have added some sentences to highlight the most important breakthroughs that are all along the main article showing the novelty of this work, such as:

- Use of graphene as a top electrode.
- XPS and operando XPS characterization in large area junctions.
- Subsequent implementation in a transistor
- PEEM and operando PEEM characterization at the microscale.
- Survey of ferrocene oxidation state in a solid junction for electronic transport discussion.
- First photodetector based on the photooxidation of ferrocene.

Line 42:

“In this work, we first investigate the properties of a single SAM//Gr hybrid junction and then introduce the same material in a new optoelectronic platform (m-GFET) based on Gr and micro self-assembled monolayers (μ SAMs) made with FcC₁₁SH. Our novel design, free of ionic liquid¹, uses Gr as a soft, high conductive and transparent top contact that encapsulates the molecular entity of interest, allowing both operando characterization and optical gating.

Line 64:

“Remarkably, Au/SC₁₁Fc//Gr SAMs continue to exhibit the oxidation-reduction process.”

Line 68:

“Another significant result is the XPS spectra of a SAM under the graphene layer.”

Line 86:

“Our graphene covering 11-(ferrocenyl)-1-undecanethiol (SC₁₁Fc) hybrid SAMs can overpass this tedious temperature dependent characterization or limitations by enabling operando spectroscopy. For example, our approach may present advantages in junctions pushed to work in the inverted Marcus Regime where the transport mechanism becomes activationless²⁸.”

Line 205:

“In order to clearly discuss the electrochemical processes occurring inside the junction, a tremendous advantage of our design is to be able to perform Photoemission electron microscopy (PEEM) following the state of the SAM while the transistor is operating.”

Lines 240:

“Finally, this section discusses the novel results on photon absorption by the Fc group and subsequent electron transfer to the Gr channel.”

1. In the title, the abstract, and the introduction, the authors emphasized the importance of reproducibility in molecular electronics and the lack of operando characterization. If the authors believed that this is the most striking part of the results, more experimental and theoretical results should be displayed, not just the XAS spectra in Figure 4d.

We thank the referee for the relevant suggestion. Regarding the reproducibility aspect we point out to the new Figure S21 and answer to referee 1 #1. Additionally, in this paper we want to present the novelty of the constructed platform, how it was achieved, from the measurements in large junctions to the m-GFET. Firstly, XPS proves how graphene reduced the aging of the SAMs and then this effect help to make, in a second step, operando PEEM measurements at ALBA synchrotron facilities. There, we have seen that the oxidation state does not vary with the source-drain voltage. Additionally, to this spectroscopic characterization we performed another series of characterization such as th-PEEM which gives us an electrical potential map of the transistor and UPS results that confirm the variation of the electrode work function, the latest relates with the observed Dirac point shift (Figure 4a). Undoubtedly these results are very interesting, but in the article production we decided to give more space to another great result such as photodetection and its potential use for joining optical and electronic circuits or fabrication of heterogeneous logic functions. This work with supplementary spectroscopic studies will be published soon as LEEM-PEEM 12 proceedings in Ultramicroscopy.

Anyway, we consider very pertinent the comment of the referee and we have now included Figure S8 with UPS results on large area Au, Au/SC₁₁Fc, and Au/SC₁₁Fc//Gr surfaces and Figure S26 with th-PEEM results on the entire m-GFET.

Figure S8. UPS spectra of: (a) Au, Au/SC₁₁Fc, and Au/SC₁₁Fc//Gr electrodes and (b) zoom at the Kinetic Energy window of 4 to 6 eV.

“Figure S26 shows the threshold PEEM measurements acquired as a function of the photo-electron energy referenced to the Fermi level, $E - E_F$. The take-off or kinetic energy is the difference between this value and the work function and it is controlled by the applied start voltage. Then for each pixel we have a map of this minimum thermodynamic work required to remove an electron from the solid to a point in the vacuum. When applying a voltage, we have an operando surface potential in our device. In Figure S26a, we observe a m-GFET under the PEEM microscopy and the selected points where start voltage spectra are taken for the different polarization bias. In Figure S26b, c and d we observe the start voltage spectra for each point at 0, 1 and -1 voltages. We can notice how the spectra are shifted, this shift is related with the voltage drop, for example between two points.”

Figure S26. Threshold PEEM. (a) mG-FET operating correctly, (b) th-PEEM spectra at $V_{SD} = 0$ V (c) th-PEEM spectra at $V_{SD} = 1$ V (d) th-PEEM spectra at $V_{SD} = -1$ V. Field of view is 50 μm . Photon energy 702 eV.

2. In fact, several figures contain the band diagrams (Fig 2ab, 4ef, 5bc) and the authors discussed the charge transport at the molecule//graphene interface. More evidence should be provided to demonstrate these mechanisms. An in-depth understanding of the unique electronic and optoelectronic transport at the interface is helpful to design molecular electronics. Chem, 2020, 6(5), 1172-1182 gives some results in this aspect. They observed a low-conductance state in the oxidized state of ferrocene with symmetrical characteristics, while a high-conductance state in the reduced state with asymmetrical curves.

We appreciate the very significant comment of the referee and take advantage of this note to extend our observations and discussion of the results. The work suggested by the referee is our reference 21, we recognize the fantastic work of C. Jia et al. and emphasize our design differences where we do not use an ionic-liquid to gate the transistor. Remarkably, reference 21 reports a RR of 2 order of magnitude for FcC₆S/Au while others in the best cases report RR of 2 orders of magnitude for FcC₁₁S/Ag and 1 order of magnitude for FcC₁₀S/Ag (our reference 14), in this sense FcC₆S/Ag would have a RR of only factor 2*.

* Yuan, L., Nerngchamng, N., Cao, L. *et al.* Controlling the direction of rectification in a molecular diode. *Nat Commun* 6, 6324 (2015). <https://doi.org/10.1038/ncomms7324>

We have updated our Figure 2 supported with our extra UPS experiments (new Figure S8) utilized to quantify the working function of Au and Au/SC₁₁Fc surfaces. In the new version, we explain the lack of rectification effect when changing the EGaIn top electrode by Gr. In principle it is reasonable to find a diode effect in a molecular junction based on Au/SC₁₁Fc//Gr since the close proximity of the Fermi level of EGaIn and Gr (4.5 eV). However, by modifying the EGaIn top electrode to Gr we observe a suppression of the rectification. Additionally, our operando XPS experiment does not detect oxidized species suggesting that the molecular diode operates in the reverse regime. We propose that for Gr top electrode, the voltage drop (η) is shifted to lower values disturbing the rectification effect.

We have modified the text accordingly with the updated Figure 2. From line 111:

~~Figure 2a and Figure 2b present forward and reverse characteristics considering a 0.7 voltage drop (η).~~ It is essential to note that the performance in the forward and reverse regimes depends on the working function of the electrodes and the voltage drop. If the molecular energy level falls between the work function of the electrodes the diode will operate in forward, otherwise in reverse. Based on the Ultraviolet photoelectron spectroscopy (UPS) (Figure S8) we obtain a working function of the decorated electrode of $\Phi_{Au/SC11Fc} = -4.5$ eV, then we consider that the only free parameter to explain the lack of rectification with the new Gr top electrode is the voltage drop (η). **Figure 2a** presents two different electron transport mechanism for η equal to 0.7 and 0.5. For a $\eta=0.7$, two

black arrows represent the sequential tunneling in the forward regime. It could explain the electrical response with the EGaIn electrode. However, a $\eta=0.5$ moves the molecular energy level away from the fermi level window of the electrodes providing a direct tunneling mechanism within the junction. That explains the electrical characteristics of the Gr top electrode, additionally our operando spectroscopy does not detect the operation in the forward regime where the contribution of the oxidized species should appear. Figure 2b presents in both cases ($\eta=0.7$ and $\eta=0.5$) a direct tunneling mechanism. ~~The onset for the two regimes is represented in.~~ The white and blue background shading in Figure 2c represents the forward and reverse regimes as function of the applied voltage and voltage drop within the junction. The orange line equals the first part of the inequality and the yellow line, the second (Equation 5 and 6 in Theoretical Methods). As shown in Figure 2c, the voltage drop within the molecular junction is a critical parameter to pay special attention. ~~Especially, for large voltage windows, there is a forward regime at positive voltages that should cause a regime with negative differential resistance.~~ “Especially, for large voltage windows, unless a carefully consideration [33], it may appear a forward producing the breakdown before reaching 2V.” Considering this approach, the lack of rectification in covered junctions may derive from the fact that Gr affects the voltage drop promoting to work in the no rectification regime (close to the black circle). Alternatively, noticing that this model is particularly sensitive to errors in the energy levels, errors of 3% may push the no rectification regime close to the 0.7 voltage drop (dashed lines).

Figure 2. Electron transport modeling at the junction. Band diagrams illustrations at (a) forward and (b) reverse bias considering $\Phi_{Gr}=4.5$ eV, $\Phi_{Au}=5.2$ eV, $E_H=5$ eV and $\eta=0.7$. (c) Operation regions for molecular diodes connected with Au and Gr electrodes as a function of the η . Dashed lines consider $\Phi_{Gr}=4.9$ eV, $\Phi_{Au}=5.4$ eV and $E_H=5$ eV. The white and blue backgrounds represent forward and reverse regimes respectively. (d) DFT calculated I(V) curve for a single molecule Au/SC₁₁Fc//Gr junction.

“**Figure 2. Electron transport modeling at the junction.** Band diagrams illustrations at (a) -1V and (b) +1V bias considering $\Phi_{Top} = -4.4$ eV, $\Phi_{Au/SC11Fc} = -4.5$ eV, $E_H = 5$ eV, $\eta = 0.7$ and $\eta = 0.5$. (c) Operation regions for molecular diodes connected with Au and Gr electrodes as a function of the η . Dashed lines consider $\Phi_{Gr} = 4.9$ eV, $\Phi_{Au} = 5.4$ eV and $E_H = 5$ eV. The white and blue backgrounds represent forward and reverse regimes respectively. (d) DFT calculated I(V) curve for a single molecule Au/SC₁₁Fc//Gr junction.”

In Figure 4e and 4f we have included the text already discussed in answer 3 to reference 2. As explained in Figure 2a we do not see a rectification effect with the Au/SC₁₁Fc//Gr junctions so we use to operate the transistors at low source to drain voltage, at the reverse regime.

Finally, our experimental results show photodetection only when we are working in the graphene electron transport regime. Thus, we are inclined to think that this mechanism is

sensitive to the number of charge carriers at that regime. Additionally, electrons can come from the ferrocene group underneath when it is excited by the incident light. We should observe (Figure 2a) that for a $\eta=0.5$ no rectification should be expected as for both polarization we fall in a forward regime for voltage larger than $\pm 1V$.

3. Some descriptions are not correct. For example, lines 181-182, “Each VBG range generates different number of charge trapped at the SiO₂//Gr interface” The interface here should be HfO₂//Gr.

We are grateful for the referee's comment and correct the description. “Each VBG range generates different number of charge trapped at the HfO₂//Gr interface”.

4. In figure 4, the authors talked about the charge traps in the device and showed the hysteresis results. However, this is not the intrinsic behavior of molecular electronic devices. The fabrication process should be optimized to minimize this effect, such as the deposition of a high-quality HfO₂ layer or the annealing. It is better to focus on the main breakthrough of this work. Some irrelevant discussions can be moved to SI.

We greatly appreciate the referee's comment. It is important to note that there are electronic mechanisms that are not directly related to the properties of the molecular junctions and it is our intention to highlight them. We therefore add the comment (Line 208): “Supra-molecular interactions, ion movement and charge trapping effects related to device geometry and fabrication conditions can alter the electronic response of the device.” However, we believe these results are important for the new platform, as we answer to referee 1 in comment 4, (Line 217): “Although in many applications trapped charges may be undesirable and attempted to minimize^{45,46}, this hysteretic mechanism may be rich for the purpose of nonvolatile memory devices and give extra performances in chemical or biological sensing.”

5. The device can respond to optical stimuli, which is quite interesting. The authors believe that light is absorbed by the molecules. The absorption spectra are absent. And a control experiment without molecules can also demonstrate it because graphene can absorb 2.3% light per layer in a broadband spectrum.

We thank the referee for his comment and correct our Figure S27 which contains our control experiment comparing a GFET and an mGFET under light pulses. Showing that GFETs without Fc functionality are not able to photo-detect.

Figure S27. Photoresponse. Drain to source conductance at dark and light pulsing conditions for (a) m-GFET and (b) GFET. Light power: 8.8 mW, $V_{DS} = 0.1$ V

REVIEWER COMMENTS

Reviewer #1 (Remarks to the Author):

The authors have answered my comments satisfactorily. I would suggest it for publication.

Reviewer #2 (Remarks to the Author):

Despite the authors' revision, many of the arguments described in the paper are still weak. For example, the authors assume the band structure of the device and explain the charge transport based on this, but additional experiments are needed to prove the authors' claim. And, the contents of the manuscript are somewhat distracting, so it is difficult to convey what kind of research content is being said. I will keep my original stance and can't recommend acceptance of this manuscript. Instead, I recommend authors to modify their manuscript and submit to other journals. Followings are comments for authors.

#1. This article consists of the first part of developing a molecular device using HfO₃ and Ferrocene, analyzing and explaining its band structure and the electrical properties, and the second part of implementing an AND/OR gate using the devices. The section "Operando charge transport in the ferrocenyl m-GFETs" and Figure 4 in the middle seem unnecessary in terms of the logical development of the text, so it may be appropriate to send it as a supplementary. Or, at least, it is necessary to revise the text so that the readers can understand why this part of text and Figure 4 are included in the main manuscript.

By the way, Table S2 can't be found in the supplementary.

And I can't understand what the authors really want to say about PEEM data.

#2. According to line 121 of the manuscript, η is referred to as "only free parameter", but the band structure must be specified in order for this to be established. First, authors need to confirm the band structure through work function measurements by experimental tools such as UPS and KPM for Au, Au/SC11Fc and Au/SC11Fc/Gr structures. In addition, although the HOMO of SC11Fc plays the most important role in the current conduction of the device, any experimental basis for its value was not presented in the manuscript. The HOMO value should be specifically specified and presented through UPS and CV measurements.

#3. Since there is no experimental basis to support Figure 2c, it is necessary to supplement it.

If $\eta = 0.5$ (or 0.7) in Figure 2c, the NDR phenomenon should appear when a voltage from -1.5 V to 1.5 V is applied to the device. Since the ferrocene/graphene structure is known to have a breakdown voltage of approximately ~1.5 V due to its robustness, this should be verified.

Furthermore, the device has a structure with gate electrode, so one controls HOMO by applying gate voltage. Then, it would be possible to make the forward region (white region) even at a low voltage through an appropriate gate voltage. This should be verified.

And, if the blue area in Figure 2c corresponds to reverse region, it corresponds to direct tunneling. Then, by measuring the current at variable temperatures, it is necessary to confirm that the charge

conduction is temperature-independent in this region. Also, the temperature-dependent charge conduction should appear outside the blue region.

By the way, paragraphs in lines 135-139 are old and should be revised. For example, in "work in the no rectification regime (close to the black circle)", there is no black circle in Figure 2c.

#4. Overall, there are several numbers or phrases in the manuscript that were just changed without being explained by the authors, such as the numbers in the band structure in Figure 4c and in Figure 2. Nevertheless, the band diagrams in Figure 4 and the band diagrams in Figure 5 still have problems, such as having different numbers written even though they have the same structure.

Considering all these, I would not recommend acceptance of this manuscript.

Reviewer #3 (Remarks to the Author):

The authors have thoroughly revised the manuscript and I believe the manuscript has been greatly improved. However, the figure capture of Figure S27 seems incorrect. Figure 27a with the name of m-GFET displays negligible photoresponse. Please double-check it and make the corresponding revision.

Firstly, we would like to thank referees #1 and #2 for their time, corrections and finally accepting the manuscript for publication.

Despite the authors' revision, many of the arguments described in the paper are still weak. For example, the authors assume the band structure of the device and explain the charge transport based on this, but additional experiments are needed to prove the authors' claim. And, the contents of the manuscript are somewhat distracting, so it is difficult to convey what kind of research content is being said. I will keep my original stance and can't recommend acceptance of this manuscript. Instead, I recommend authors to modify their manuscript and submit to other journals. Followings are comments for authors.

We acknowledge the time and comments of referee #2. In fact, we are especially pleased that all their previous concerns as the importance of Figure 5, demonstration of a heterogeneous logic gate and easy realization of a logic gate with only electrical inputs have been understood. However, we are sorry that **new doubts** have emerged.

Regarding the referee's opinion on this research and the number of experiments, it seems a subjective opinion. In fact, it is possible that other groups may find this design motivating and want to reproduce and expand the results presented here. In this article they will find how to make it, some interesting properties and possible applications.

Even if the referee finds the investigation distracting, we would like to emphasize that our experimental design is extremely simple. It consists on testing a hybrid component in large molecular junctions and then exploit the same material in the fabrication of a transistor to explore both **operando characterization** (Figure 4) and **optical gating** (Figure 5).

#1. This article consists of the first part of developing a molecular device using HfO₃ and Ferrocene, analyzing and explaining its band structure and the electrical properties, and the second part of implementing an AND/OR gate using the devices. The section “Operando charge transport in the ferrocenyl m-GFETs” and Figure 4 in the middle seem unnecessary in terms of the logical development of the text, so it may be appropriate to send it as a supplementary. Or, at least, it is necessary to revise the text so that the readers can understand why this part of text and Figure 4 are included in the main manuscript.

By the way, Table S2 can't be found in the supplementary.

And I can't understand what the authors really want to say about PEEM data.

We do not understand how in the previous paragraph referee #2 indicates that we assume the band structure and here exposes that we analyze and explore it. Our device use HfO₂ to passivate the gold electrodes from graphene and precisely determine the molecular decorated areas.

Figure 4 is the demonstration of the in-operando spectroscopy. **A claim already set in the title.**

We acknowledge the comment on table S2. The sentence in the main article has been modified to: (Calculated with equation 8 in Theoretical Methods).

With PEEM data and when applying a voltage, we obtain an operando surface potential. For example, we take advantage of it to detect any leaks or malfunctions. Alternatively, considering the referee's interest in the band structure of the device, PEEM can operate in ARPES-mode (k-space) to probe the allowed energies and momenta of the electrons in our device.

#2. According to line 121 of the manuscript, η is referred to as “only free parameter”, but the band structure must be specified in order for this to be established. First, authors need to confirm the band structure through work function measurements by experimental tools such as UPS and KPM for Au, Au/SC11Fc and Au/SC11Fc/Gr structures. In addition, although the HOMO of SC11Fc plays the most important role in the current conduction of the device, any experimental basis for its value was not presented in the manuscript. The HOMO value should be specifically specified and presented through UPS and CV measurements.

UPS experiments can be found in figure S8, those results determine the work function and establish the numbers in the band structure specified in figure 2.

CV experiments are presented in figure 1 and figure S1. HOMO is calculated with equation 2 in Theoretical Methods as indicated in line 60.

With the results of these experiments, we understand that the only parameter we are not able to determine is the potential drop at the junction.

Furthermore, the data obtained by UPS in Figure S8, which show a change in the work function between the non-functionalized and SAM-functionalized gold electrodes of 0.5 eV correlates perfectly with the shift observed in Figure 4a. Actually, **figure 4 is helping to understand the Band Structure of the device.** To make this link stronger we add in line 194: in agreement to our UPS experiments (Figure S8).

#3. Since there is no experimental basis to support Figure 2c, it is necessary to supplement it.

As indicated in the previous comment UPS and CV experiment support Figure 2c. We add in line 148: (Data from UPS experiments in Figure S8).

If $\eta = 0.5$ (or 0.7) in Figure 2c, the NDR phenomenon should appear when a voltage from -1.5 V to 1.5 V is applied to the device. Since the ferrocene/graphene structure is known to have a breakdown voltage of approximately ~ 1.5 V due to its robustness, this should be verified.

After our previous discussion, NDR property is **not discussed in the article anymore**. Additionally, some of our references explore the breakdown voltage, that again is not the objective in our study. We only highlight the work of those references.

Furthermore, the device has a structure with gate electrode, so one controls HOMO by applying gate voltage. Then, it would be possible to make the forward region (white region) even at a low voltage through an appropriate gate voltage. This should be verified. And, if the blue area in Figure 2c corresponds to reverse region, it corresponds to direct tunneling. Then, by measuring the current at variable temperatures, it is necessary to confirm that the charge conduction is temperature-independent in this region. Also, the temperature-dependent charge conduction should appear outside the blue region.

This hypothesis, notably not mentioned in the first review, could be the subject of a fascinating new article. In fact, we disagree with this hypothesis. A correct understanding of Figure 4b suggests that it is not possible to access the HOMO by controlling the back gate. This fact would produce a jump in the curves presented in Figure 4b just as it occurs when light is applied, Figure 5a.

By the way, paragraphs in lines 135-139 are old and should be revised. For example, in “work in the no rectification regime (close to the black circle)”, there is no black circle in Figure 2c.

Certainly, these sentences have been removed.

#4. Overall, there are several numbers or phrases in the manuscript that were just changed without being explained by the authors, such as the numbers in the band structure in Figure 4c and in Figure 2. Nevertheless, the band diagrams in Figure 4 and the band diagrams in Figure 5 still have problems, such as having different numbers written even though they have the same structure.

“These numbers” refers to the UPS and CV mentioned the text (lines 115 and 60).

Considering all these, I would not recommend acceptance of this manuscript.

We sincerely appreciate the referee's review, even though our feeling has been a little confused since: (1) there are new criticisms that did not appear in the first review, (2) referee #2 makes mention of electrical properties that no longer appear in the manuscript (NDR), (3) referee #2 does not find experimental results such as voltammetry (figure 1 and figure S1) or UPS (figure S8) and finally, (4) referee #2 proposes to perform new experiments that may be the subject of a new and very specific article. We hope that this point-by-point answer will help to understand any doubt about our research.

REVIEWERS' COMMENTS

Reviewer #1 (Remarks to the Author):

The authors have answered my comments thoroughly. Regarding the authors' response to the comments of Ref #2, I would like to suggest the authors to seriously answer the questions of Ref. #2, such as provide Table S2 in the SI in addition to citing the reference, and make appropriate revisions or discussion about PEEM data and the NDR phenomenon, which should be highlighted in the main text. The manuscript should be publishable, depending on the suitable revisions.

Reviewer #3 (Remarks to the Author):

I feel that the authors have provided sufficient experimental evidence to support the proposed mechanisms at the basis of their device operation. However, I urge the authors to carefully read and revise the typos in the manuscript and in the figures. I believe that these typos create huge difficulty for the referees to understand this work and it will easily lead to the misunderstanding. For example, in Figure 2a and 2b, Au/SC11Fc is written as Au/S11Fc.

Firstly, we thank referees #1 and #2 for all their dedication, honesty, time, corrections, guiding and finally accepting the manuscript for publication.

Reviewer #1 (Remarks to the Author):

The authors have answered my comments thoroughly. Regarding the authors' response to the comments of Ref #2, I would like to suggest the authors to seriously answer the questions of Ref. #2, such as provide Table S2 in the SI in addition to citing the reference, and make appropriate revisions or discussion about PEEM data and the NDR phenomenon, which should be highlighted in the main text. The manuscript should be publishable, depending on the suitable revisions.

We have now updated line 209 in the main text: "The number of trapped charges (N) are calculated with Equation 8 in Methods and displayed in Supplementary Table 2." In Supplementary Table 2 we summarized the calculated number of charges trapped for experiments presented in Figure 4b. Several V_{BG} windows produce different V_{BG} and related trapped charges that we calculate with Equation 8 in Methods.

Regarding the threshold PEEM discussion we have added in the main text (line 238): "Additionally, threshold PEEM measurements were performed to obtain a surface potential map of the m-GFETs. In Supplementary Figure 26 can be seen how the spectra are shifted when applying different V_{DS} (-1,0 and +1V), this shift is related to the voltage drop." An extended discussion could be found in Supplementary Note 9.

Finally, NDR phenomenon is not mentioned in the article. We removed our sentence after the first revision because we understood that our reasoning could be exaggerated or misinterpreted. Moreover, it was not part of the main results. Nonetheless, NDR is undoubtedly an interesting phenomenon that can be investigated in future works with this platform.

Reviewer #3 (Remarks to the Author):

I feel that the authors have provided sufficient experimental evidence to support the proposed mechanisms at the basis of their device operation. However, I urge the authors to carefully read and revise the typos in the manuscript and in the figures. I believe that these typos create huge difficulty for the referees to understand this work and it will easily lead to the misunderstanding. For example, in Figure 2a and 2b, Au/SC11Fc is written as Au/S11Fc.

We thank the comments and great work of the referee, understanding the article despite the difficulties that may arise in these processes. We are committed to review these typographical errors.